# Multi-Sensor-Based Occupancy Prediction in a Multi-Zone Office Building with Transformer

**Irfan Qaisar** †, **Kailai Sun** †, **Qianchuan Zhao** *, **Tian Xing** and **Hu Yan**

Center for Intelligent and Networked Systems, Department of Automation, BNRist, Tsinghua University, Beijing 100084, China; irfan21@mails.tsinghua.edu.cn (I.Q.); 18813126518@163.com (K.S.); xingt19@mails.tsinghua.edu.cn (T.X.); yanh20@mails.tsinghua.edu.cn (H.Y.)
* Correspondence: zhaoqc@mail.tsinghua.edu.cn
† These authors contributed equally to this work.

**Abstract:** Buildings are responsible for approximately 40% of the world's energy consumption and 36% of the total carbon dioxide emissions. Building occupancy is essential, enabling occupant-centric control for zero emissions and decarbonization. Although existing machine learning and deep learning methods for building occupancy prediction have made notable progress, their analyses remain limited when applied to complex real-world scenarios. Moreover, there is a high expectation for Transformer algorithms to predict building occupancy accurately. Therefore, this paper presents an occupancy prediction Transformer network (OPTnet). We fused and fed multi-sensor data (building occupancy, indoor environmental conditions, HVAC operations) into a Transformer model to forecast the future occupancy presence in multiple zones. We performed experimental analyses and compared it to different occupancy prediction methods (e.g., decision tree, long short-term memory networks, multi-layer perceptron) and diverse time horizons (1, 2, 3, 5, 10, 20, 30 min). Performance metrics (e.g., accuracy and mean squared error) were employed to evaluate the effectiveness of the prediction algorithms. Our OPTnet method achieved superior performance on our experimental two-week data compared to existing methods. The improved performance indicates its potential to enhance HVAC control systems and energy optimization strategies.

**Keywords:** occupancy prediction; deep learning; multi-sensor fusion; Transformer





## 1. Introduction

In recent years, the world has witnessed a significant surge in demand for energy, which can be attributed to a combination of factors, including a growing global population and rising living standards [1]. While these advancements have undoubtedly enhanced the quality of life, they have substantially increased carbon emissions. This rise in emissions is a significant contributor to climate change, resulting in severe consequences like extreme weather events, alterations in weather patterns, and global warming [2]. As the global population continues to grow, energy demand is expected to escalate further, posing a considerable challenge in balancing energy requirements with environmental sustainability. Therefore, enhancing energy efficiency is a widely discussed topic in energy. The focus is on discovering methods enabling use of less energy while still maintaining or improving the quality of services provided. This concept applies to various sectors, including buildings, transportation, and industry, where systems are designed with energy efficiency in mind [3]. By striving for optimal energy efficiency in these sectors, significant economic benefits can be achieved by reducing the expenses associated with energy consumption.

Buildings are responsible for approximately 40% of the world's energy consumption and release around 36% of the total carbon emissions [4]. Within building systems, the heating, ventilation, and air conditioning (HVAC) system is the most considerable portion, accounting for 40% of the energy usage [5]. Notably, the HVAC system plays a vital role in

regulating indoor temperature [6–8], ensuring thermal comfort [9], and enhancing indoor air quality (IAQ) [10,11]. In this sense, it is crucial to emphasize the significance of building performance evaluation (BPE) [12]. Buildings are carefully designed to provide occupants with a pleasant and comfortable environment [13]. However, it is essential to recognize that the number and presence of occupants can directly impact energy consumption [14]. Various studies have explored the connection between occupants' behavior and energy consumption, aiming to understanding their impact on energy consumption. In [15], a scientific approach was adopted to quantify occupants' behavior consistently. This approach encompassed factors such as occupants' presence, movement, and interactions with the energy systems installed in buildings. The objective was to integrate these behavioral aspects into building performance simulation programs for a comprehensive analysis. A detailed review conducted by [16] highlighted the dire need for adaptation to occupancy variation; the studies reviewed showed the primary sources of inefficiency in the building system, namely, irregular and partial occupancy. Prior research has demonstrated significant reductions in building energy usage by aligning HVAC systems with actual occupancy patterns [17,18]. For example, the simulation results presented in [19], showed that there was the potential to achieve energy savings ranging from 11% to 34% across different climatic regions while maintaining occupant comfort levels.

The review article [20] on occupancy prediction research highlighted the existence of two distinct categories: "occupancy detection/estimation" [21,22] and "occupancy forecast". To understand these categories, it is essential to consider the concept of the prediction window. In the context of occupancy prediction, "occupancy detection" pertains to predicting the occupancy for the current time step, providing real-time information about the current occupancy state. On the other hand, "occupancy forecast" involves predicting the occupancy for a future time step, enabling insights into future occupancy patterns. The research on occupancy prediction specifically focusing on forecasting occupancy for future time windows needs to be improved and is often conflated with detection methods. As a result, the value of occupancy forecasting may need to be emphasised or mixed with real-time occupancy detection.

Predicting future occupancy has promise for facilitating building operations and energy efficiency [23]. Nevertheless, accurately predicting occupancy is complex due to the stochastic nature of occupant presence and the inherent variability in individual behavior [24,25]. The ongoing advancements in sensor technologies, data analytics, and prediction algorithms offer promising avenues for enhancing the accuracy and reliability of future occupancy predictions. In the past decade, there have been substantial advancements in forecast algorithms, leading to significant improvements in the accuracy of occupancy predictions. These forecast methods can be categorized into four main groups [20]: conventional statistical approaches (i.e., Markov-chain-based [26] and recursive models [27]), unsupervised machine learning approaches (i.e., k-means, k-nearest neighbor techniques, and support vector clustering [28,29]), supervised machine learning approaches (i.e., gradient boosting [30], support vector regression [31], decision tree [32], random forest [33], and deep neural networks [33]), and hybrid approaches. Each group employs different techniques and methodologies to forecast occupancy patterns and behaviors.

In the literature, predicting occupancy has been an area of significant interest for researchers. However, a common trend in this field is that most researchers rely on single sources to collect input data [20]. While this approach may seem practical and convenient, it carries inherent risks. If a sensor fails, the collected data may omit valuable information, resulting in incomplete and potentially unreliable predictions. On the other hand, every source of data collection has its limitations. For example, PIR occupancy sensors are widely used in lighting controls, but they have limitations in detecting stationary occupants [34]. $CO_2$-based approaches have constraints such as low sensitivity to occupant mobility and slow response to drastic occupancy changes [35]. WiFi-based monitoring systems may also face challenges, such as connection problems, limited battery life of connected devices, and poor connections for accurate occupancy detection in large-scale buildings with many

occupants [36]. These factors can significantly affect the performance of the data collection approaches. Therefore, a better solution is to utilize multi-source data collection approaches. By adopting this approach, researchers can reduce the risk of missing out on valuable information in the event one source fails. This strategy is especially critical in occupancy prediction situations where the presented information carries significant weight and is of utmost importance. Researchers can use multiple sources to ensure their predictions are accurate, reliable, and comprehensive. However, the multi-source data for occupancy prediction needs to be improved in buildings.

To address the above problems, we introduce an occupancy prediction Transformer network (OPTnet) for building occupancy prediction. It can robustly predict occupancy presence in diverse rooms and time horizons. We fuse and feed multi-sensor data into a Transformer model to obtain the future occupancy presence in multiple zones. We provide experimental analysis and comparison between existing occupancy prediction methods and diverse time horizons. The main contributions of this paper are as follows:

- We introduce OPTnet, a Transformer-based multi-sensor building occupancy prediction network to learn an effective fused representation.
- We process two-week real operating sensor data from a multi-zone office building to predict accurate occupancy, including building occupancy, indoor environmental conditions, and HVAC operations.
- Through experimental analysis and comparison, we found that the OPTnet method outperformed existing algorithms (e.g., decision tree (DT), long short-term memory networks (LSTM), multi-layer perceptron (MLP)).
- Considering long or short occupancy prediction applications, we provide a comprehensive analysis and comparison of diverse time horizons to highlight the importance of choosing the suitable time horizon.

## 2. Methods

This research aims to predict occupancy using a dataset from our previous work [37]. The dataset was collected in an office building in Hebei Province, China, from 9 August to 21 August 2021, spanning two weeks. Further details regarding the dataset can be found in Section 3.

The research framework in this study consisted of four key steps. Firstly, sensor data and corresponding occupant information were collected from the office building, serving as the ground truth for occupancy. Secondly, data normalization techniques were applied to standardize all features to a single scale or range. Thirdly, occupancy prediction was performed using OPTnet, and its performance was compared with various machine learning algorithms, including DT, LSTM, and MLP. Lastly, performance evaluation metrics were employed to assess the accuracy and effectiveness of the prediction algorithms. The comprehensive methodology, which includes each step, is thoroughly explained in the following subsections and visually illustrated in Figure 1.

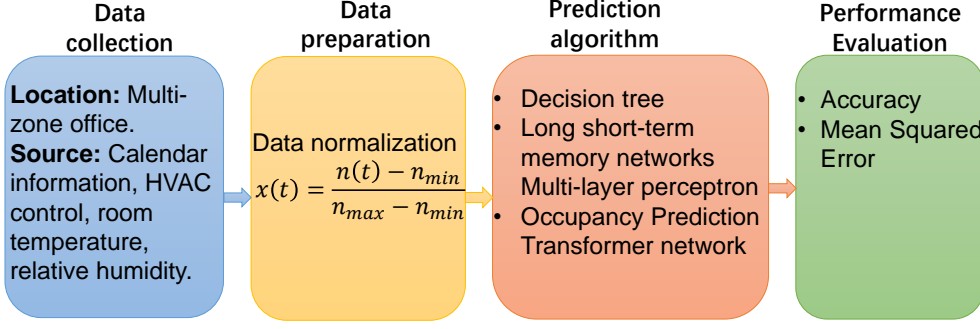

**Figure 1.** Illustration of the step-by-step methodology for occupancy prediction in buildings.

## 2.1. Data Collection

The dataset employed in this study was constructed by gathering 829,440 data points derived from the system's operational data from 9 August to 21 August 2021. In order to make sure the dataset was relevant, we filtered it to only include times when the HVAC system was on, specifically on weekdays (Monday through Friday) from 9:00 a.m. to 7:00 p.m. Consequently, the dataset encompassed data collected at a resolution of 1 min during the operational hours of the HVAC system. The dataset spanned two weeks: 9–13 August (week 1) and 16–20 August (week 2), amounting to 10 days. A total of 600 data samples were recorded daily, comprising 54 numerical values (6 zones and 9 features). Therefore, the complete dataset encompassed 324,000 numerical values ($10 \times 600 \times 54$).

## 2.2. Data Preparation

The dataset utilized in the system consisted of raw data with numerical values spanning diverse ranges. For instance, the indoor temperature across each zone fluctuated between 22 and 32 °C, the number of occupants varied from 0 to 10, and the control signal for the FCUs (fan coil units) ranged from 0 to 3.

Such disparate distributions in the raw data introduce complexity during the training process. In order to tackle this issue, data normalization techniques were utilized. By normalizing the data, the possibility of gradient explosion in deep learning is minimized, which speeds up convergence, stabilizes training, and improves the model's overall performance [7]. Prior to inputting the data into the algorithms, all raw data underwent normalization through the following steps:

$$x(t) = \frac{n(t) - n_{min}}{n_{max} - n_{min}},$$ (1)

where $n(t)$ denotes the true number of occupants at time $t$, while $n_{min}$ and $n_{max}$, respectively, denote the minimum and maximum number of occupants.

Furthermore, when the prediction algorithms generated the predicted occupancy, the output value was converted to the corresponding number of occupants through the following transformation:

$$\bar{n}(t) = \bar{x}(t) * (n_{max} - n_{min}) + n_{min},$$ (2)

where the predicted value of algorithms is denoted as $\bar{x}(t)$, while the predicted number of occupants is represented by $\bar{n}(t)$.

## 2.3. Algorithms

This research paper presents OPTnet, designed explicitly for building occupancy prediction. Furthermore, a comprehensive comparative analysis was conducted to assess the performance of this algorithm in comparison to established machine learning techniques, including DT, LSTM, and MLP. The subsequent sections of the paper provide detailed explanations and insights into these algorithms' underlying principles and operational mechanisms.

### 2.3.1. Decision Tree

DT models categorize and generalize datasets into predefined data analysis and machine learning classes. The primary objective of a decision tree is to construct a classification model that can predict the value of a target attribute (response) based on multiple input attributes (predictors). Each internal node or leaf node within the decision tree corresponds to one of the predictors, and the number of branches emerging from a categorical internal node (leaf node) is equivalent to the possible values of the associated predictor. The leaf nodes represent specific values of the response variable and are reached by traversing the path from the root node, which is the starting point of the tree, to the final leaf (possible answers) [32].

### 2.3.2. Long Short-Term Memory Networks

The LSTM network is a specialized variant of recurrent neural networks (RNNs) developed in 1997 [38]. Designed to address the challenges posed by vanishing and exploding gradients in standard RNNs, LSTM networks leverage the back-propagation through time (BPTT) algorithm to train and excel in tasks involving long-term dependencies. In contrast to conventional neuron-based architectures, LSTM networks feature memory blocks consisting of memory cell units capable of retaining state values over extended periods. Moreover, these memory blocks incorporate three distinct gate units responsible for learning how to preserve, utilize, or discard states as needed. The connectivity between memory blocks is established through layers, facilitating the overall functionality and effectiveness of LSTM networks [39].

### 2.3.3. Multi-Layer Perceptron

The MLP is a feed-forward artificial neural network (ANN) that draws inspiration from the functioning of the human brain [40]. This network comprises at least three layers of neurons, specifically the input, hidden, and output layers. MLP can effectively capture non-linear relationships between predictor variables and labels by employing activation functions, except for the input layer. In this study, the rectified linear unit (ReLU) activation function is implemented in the hidden layers as it is commonly recommended for developing neural networks [41]. Additionally, linear and sigmoid functions are adopted in the output layers for regression and classification models. To facilitate the learning process, backpropagation, a supervised learning technique, determines the optimal weights and bias values for each neuron.

### 2.3.4. Occupancy Prediction Transformer Network

With the rapid development of Chatgpt and visual foundation models, the Transformer has become a state-of-the-art (SOTA) deep learning method. Transformer, a neural network, is adequate for dealing with sequence-to-sequence (seq2seq) tasks and learning a deep understanding of sequential data. Inspired by RNNs, the Transformer follows the encoder-decoder architecture to learn aggregated hidden-layer features. Unlike RNNs, the Transformer does not perform data processing in sequential order but processes the sequential input data in parallel. In particular, encoders and decoders, composed of multiple self-attention layers, are stacked to extract multi-layer features. Multi-head attention mechanisms are applied to learn the correlation between tokens.

$$Attention(Q, K, V) = softmax(\frac{QK^T}{\sqrt{d_k}})V \tag{3}$$

In this paper, we develop an OPTnet for the occupancy prediction model. We formulate the occupancy prediction as a sequence prediction problem. In Figure 2, we treat the history occupancy and environmental factors as the OPTnet's inputs while treating future occupancy information (presence or number) as the OPTnet's outputs. The structure OPTnet is shown in Figure 2.

### 2.4. Performance Evaluation

We define two evaluation indicators for occupancy prediction as follows:

$$\text{MSE} = \frac{1}{N} \sum_{f=1}^{N} (n(t) - \bar{n}(t))^2,$$

$$\text{Accuracy} = \frac{1}{N} \sum_{t=1}^{N} (1 - \text{sign}|n(t) - \bar{n}(t)|), \tag{4}$$

where $n(t)$ is the true number of occupants in the time $t$, and $\bar{n}(t)$ represents the predicted number of occupants. $N$ is the time length. The mean squared error (MSE) indicates the

difference between the predicted value and the ground truth, while the accuracy indicates the hit rate of occupancy prediction. The performance is better when the MSE is smaller and the accuracy is bigger.

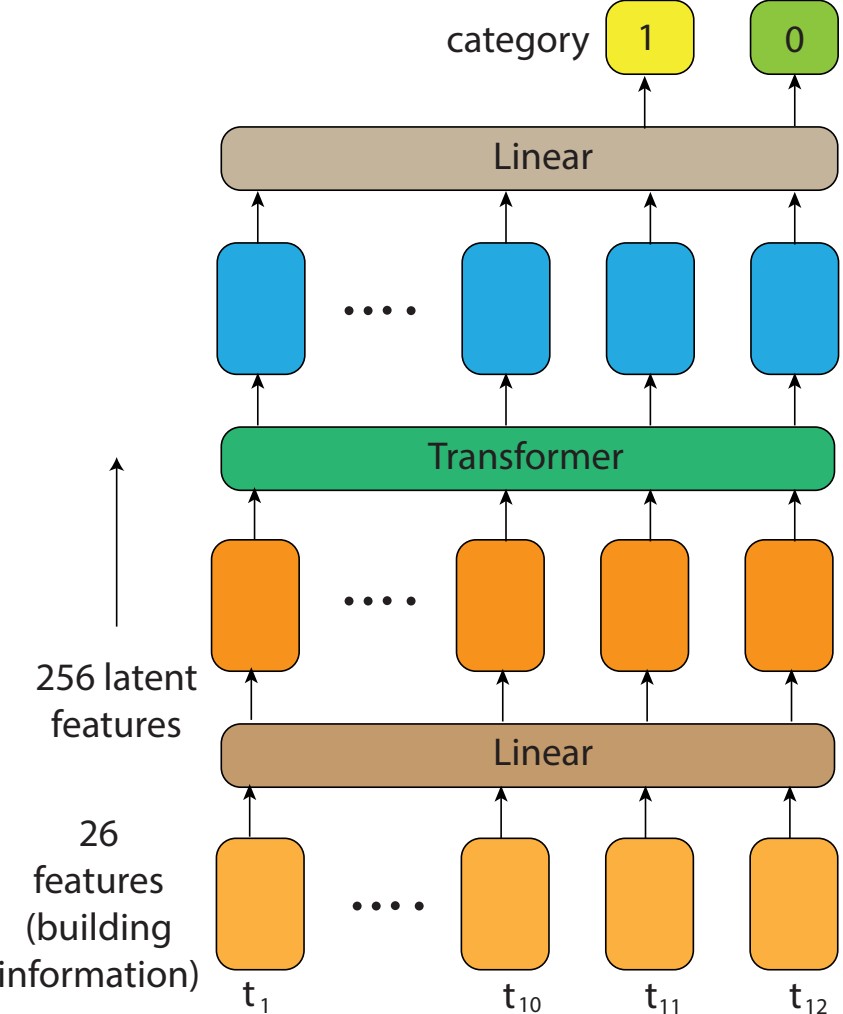

**Figure 2.** The structure of OPTnet.

## 3. Experiments

### 3.1. Experimental Environment

Our experimental system in Hebei, China, represents a multi-zone office environment. It comprises seven distinct working zones, a refrigeration station, and an activity room. Within the system, we employ various components to facilitate efficient operation. These include an air source heat pump (HP) for cooling purposes, variable frequency water pumps for circulation, a fan coil unit (FCU) for indoor HVAC control, and cameras for video capture to monitor the environment.

The building is divided into nine regions: an activity room, a refrigeration station, and seven working zones. Zones 4 to 6 are virtually separated from a more extensive zone, following the specifications of the IoT system deployment [37]. Figure 3 provides an overview of the entire office building system, showcasing the arrangement and functionalities of each zone. It is important to note that data from Zone 3, which is the financial office, is not publicly accessible. Therefore, our experiments focus on the remaining zones (1, 2, and 4–7).

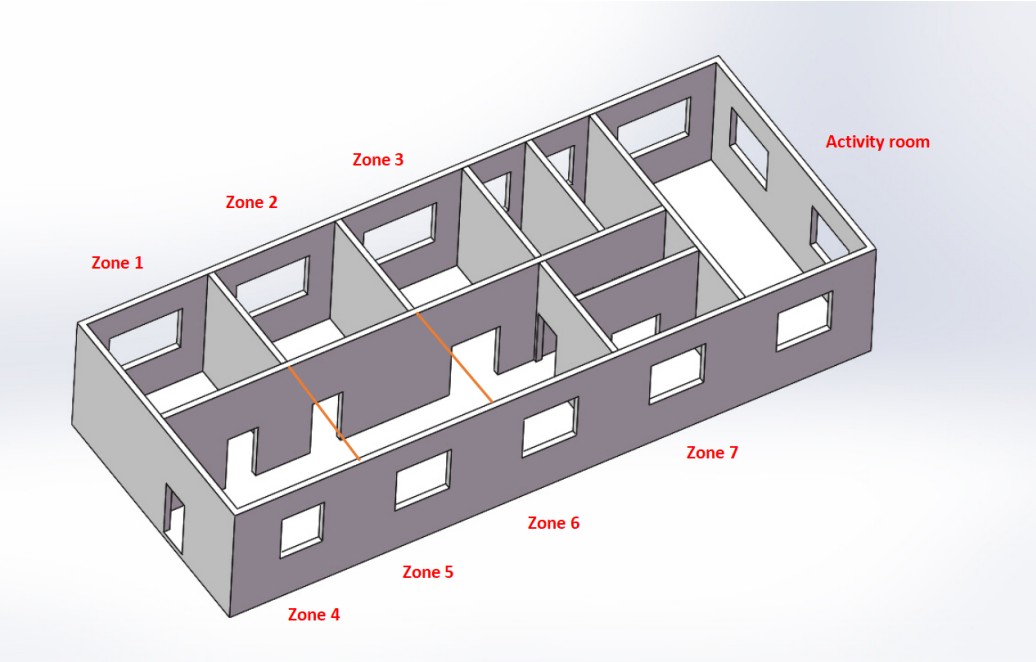

**Figure 3.** The experimental multi-zone office [37].

Each zone has an indoor temperature sensor (ITS), a device management panel for controlling the FCU, a video recording camera, and a Jetson Nano for video processing and zone occupancy estimation. The system also contains an outdoor temperature sensor (OTS), which is deployed on the north external wall, for collecting the outdoor data. Please refer to our previous work in [37] for more detailed information on the experimental system and IoT architectures.

*3.2. Experimental Data*

Given the substantial influence of office occupants on the energy performance of an office, the experimental data for the occupancy prediction model is based on the routines and behaviors of these occupants. These data are categorized into four groups: calendar, occupancy, indoor environment, and HVAC control.

- The calendar information: We collected the sensor data from 9:00 to 19:00 during the five working days (from Monday to Friday) and weekends (Saturday and Sunday).
- The occupancy information: We captured videos from our cameras. Then, we analyzed and estimated occupancy presence (1 or 0) in each room using advanced artificial intelligence technologies. The time resolution was 1 min. The duty ratios of occupancy in multi-zones are shown in Table 1. The duty ratios are various, indicating that the practical dataset is diverse and complete.
- The indoor environment information: We collected the indoor temperature and relative humidity data, directly affecting the occupants' thermal comfort. The temperature and relative humidity data can be used to predict future occupancy.
- The HVAC control information: The HVAC system employs FCUs for control. The control signs (FCU temperature feedback, FCU control mode, FCU on/off feedback, and FCU fan feedback) are considered for occupancy prediction.

**Table 1.** Duty ratios of occupancy in multi-zones.

| Weeks | Zone 1 | Zone 2 | Zone 4 | Zone 5 | Zone 6 | Zone 7 |
|--------|--------|--------|--------|--------|--------|--------|
| Week 1 | 0.314 | 0.504 | 0.919 | 0.405 | 0.898 | 0.611 |
| Week 2 | 0.084 | 0.854 | 0.816 | 0.353 | 0.898 | 0.256 |

### 3.3. Experimental Parameters

A recommendation in [23] emphasizes the importance and effectiveness of occupancy prediction models for occupancy-based HVAC control systems.

In our experiments, We chose the historical multi-sensor data (including occupancy presence, HVAC control, FCU temp feedback, FCU control mode, FCU on-off feedback, FCU fan feedback, room temperature 1 and 2, room relative humidity 1 and 2) as the method inputs. We chose occupancy presence (1 or 0) as the method outputs. We compared well-known occupancy prediction methods (DT, LSTM, MLP) and OPTnet. To compare LSTM and OPTnet reasonably, we fixed the hyperparameters:

- The Adam optimizer trained the LSTM and OPTnet model for 20 epochs.
- The learning rate was $10^{-4}$.
- The batch size was 4.
- The numbers of LSTM and TOPTnet layers were 6.
- The number of fully connected layers was 5.
- The dropout of the last layer was 0.5.
- MSE loss function.

Considering long or short prediction applications, we used the 30 min historical multi-sensor data to predict occupancy presence in diverse time horizons. In other words, we compared the occupancy prediction performance with 1 min, 2 min, 5 min, 10 min, 20 min, and 30 min horizons.

## 4. Results and Discussion

The DT, LSTM, MLP, and OPTnet algorithms were implemented on a dataset obtained from a multi-zone office building. The performance of these algorithms with different time horizons was evaluated by measuring their accuracy and MSE values for each zone. The results are presented in Tables 2 and 3, respectively, showcasing the accuracy and the corresponding MSE.

### 4.1. OPTnet vs. (LSTM, MLP, DT)

In Table 2, it is evident that the OPTnet exhibited exceptional accuracy values for both weeks in Zones 1 and 4. Across various time horizons, the OPTnet consistently outperformed the other machine learning algorithms, demonstrating its superior predictive capabilities. Moving to Zone 2, the accuracy values of the DT algorithm were consistently perfect (the accuracy was 1) for week 1 but comparatively lower for week 2 compared to the other machine learning algorithms. However, the Transformer algorithm showed high and consistent accuracy values for week 1 across different time horizons, and even higher accuracy values for week 2, surpassing the performance of the other machine learning algorithms.

In Zone 5, the OPTnet achieved high accuracy values for both weeks, particularly for smaller time horizons such as 1 min and 2 min. However, as the time horizon increased, the accuracy of the OPTnet decreased. Shifting to Zone 6, the accuracy values of the MLP algorithm were consistently higher than for the other machine learning algorithms for different time horizons in both weeks. Lastly, in Zone 7, the OPTnet demonstrated higher accuracy values than the other machine learning algorithms for week 1, while the MLP algorithm outperformed other algorithms for week 2.

Upon careful analysis of the results, specifically the accuracy and MSE values for each zone, it was evident that the OPTnet mostly outperformed the other machine learning algorithms with regard to the diverse duty ratios in Table 1. Here are some reasons why the OPTnet outperformed these methods:

- Occupancy patterns in buildings can exhibit long-range dependencies, where the presence or absence of occupants in one area can impact occupancy in other areas. The self-attention mechanism in the OPTnet allows it to capture such long-range

dependencies effectively. In contrast, DT, LSTM, and MLP struggle to model these dependencies explicitly.

- Occupancy patterns often have temporal dynamics, where the presence or absence of occupants at one time influences future occupancy. The OPTnet, with its self-attention mechanism, can capture these temporal dynamics by attending to relevant past occupancy information at each time step. On the other hand, DT typically considers each time step independently, LSTM focuses on short-term dependencies, and MLP lacks inherent mechanisms for capturing temporal dynamics.
- OPTnet can use parallel computation, making it highly scalable and efficient, especially when dealing with large datasets. This scalability allows the Transformer model to handle complex occupancy prediction tasks efficiently. In comparison, DT, LSTM, and MLP may need to improve scalability and computational efficiency, mainly when dealing with longer sequences or large datasets.
- OPTnet has shown robustness to noisy data due to its ability to attend to relevant information and suppress noise during the attention mechanism. This robustness can benefit occupancy prediction tasks, where the data may contain missing or noisy observations. DT, LSTM, and MLP are more sensitive to noisy data and require additional preprocessing or regularization techniques to handle such scenarios.

The superior performance of the OPTnet highlights its effectiveness in accurately predicting occupancy patterns within different building zones. This outcome underscores the significance of utilizing the OPTnet as a reliable and robust approach for occupancy prediction in diverse environments. The improved performance of the OPTnet signifies its potential to enhance the efficiency and effectiveness of various applications that rely on accurate occupancy forecasts, such as HVAC control systems, energy optimization strategies, and overall building management.

### 4.2. Time Horizons vs. Performance

We noticed a clear pattern after analyzing the performance of OPTnet and the other machine learning algorithms across different time horizons. **As the time horizon became longer, the accuracy of each algorithm tended to decrease while the MSE value tended to increase.** This finding highlights the importance of selecting an appropriate time horizon based on the specific application requirements. The effect of the time horizon on algorithm performance highlights the importance of choosing the right time window for different purposes. Short-term time horizons are beneficial for applications needing instant occupancy predictions or real-time monitoring. This enables better capture and response to short-term occupancy changes with higher accuracy. On the other hand, longer time horizons are better for applications that focus on long-term occupancy forecasting and trend analysis. Even though the accuracy may be slightly lower, having a broader view of occupancy patterns and trends over a more extended period is valuable for tasks like energy planning, resource allocation, and managing occupancy in the long run.

There are two exceptions to the point that accuracy decreases with increasing horizons. In Zones 4 and 6, we found the accuracy was increasing. We noticed that, as shown in Table 1, the duty ratios of occupancy in Zone 4 and 6 were very high (0.919, 0.816, 0.898, 0.898). With increasing horizons (1, 2, 5, 10, 20, 30 min), in the historical occupancy data, the weight of occupancy presence became bigger while the weight of occupancy absence became smaller. Thus, the OPTnet and machine learning algorithms became more conservative to achieve high accuracy, even though we used balanced class weights to train our models. When the time horizon grows, conventional models converge to the class with the most duty ratios and predict occupancy presence. This is a limitation of our proposed framework.

**Table 2.** The accuracy evaluation for zones 1, 2, 4–7. Bold means better performance.

| Time Horizon | Week-1 | | | | Week-2 | | | |
|---|---|---|---|---|---|---|---|---|
| | DT | LSTM | MLP | OPTnet | DT | LSTM | MLP | OPTnet |
| **Zone 1** | | | | | | | | |
| 1 min | 0.818 | 0.949 | 0.686 | **0.951** | 0.854 | 0.961 | 0.916 | **0.967** |
| 2 min | 0.782 | **0.923** | 0.909 | 0.915 | 0.928 | 0.942 | **0.945** | 0.931 |
| 5 min | 0.793 | **0.821** | 0.812 | **0.82** | 0.861 | **0.92** | 0.911 | **0.92** |
| 10 min | 0.772 | 0.682 | 0.596 | **0.774** | 0.798 | 0.901 | 0.912 | **0.922** |
| 20 min | 0.701 | 0.622 | 0.478 | **0.716** | 0.825 | 0.824 | 0.767 | **0.857** |
| 30 min | 0.665 | 0.585 | 0.549 | **0.676** | **0.851** | 0.654 | 0.55 | 0.827 |
| **Zone 2** | | | | | | | | |
| 1 min | **1** | 0.798 | 0.504 | 0.751 | 0.777 | 0.865 | 0.854 | 0.863 |
| 2 min | **1** | 0.703 | 0.847 | 0.768 | 0.746 | 0.852 | **0.86** | 0.854 |
| 5 min | **1** | 0.679 | 0.814 | 0.773 | 0.772 | 0.829 | 0.834 | **0.854** |
| 10 min | **1** | 0.497 | 0.596 | 0.782 | 0.754 | 0.852 | 0.818 | **0.853** |
| 20 min | **0.984** | 0.498 | 0.787 | 0.714 | 0.736 | **0.849** | 0.839 | **0.849** |
| 30 min | **0.953** | 0.499 | 0.803 | 0.778 | 0.733 | **0.847** | 0.818 | **0.847** |
| **Zone 4** | | | | | | | | |
| 1 min | 0.893 | **0.919** | **0.919** | **0.919** | 0.718 | 0.816 | 0.816 | **0.9** |
| 2 min | 0.912 | 0.92 | 0.92 | **0.952** | 0.774 | 0.815 | 0.815 | **0.837** |
| 5 min | 0.899 | 0.938 | **0.923** | **0.923** | **0.864** | 0.814 | 0.855 | 0.817 |
| 10 min | 0.917 | **0.927** | **0.927** | **0.927** | **0.821** | 0.819 | 0.819 | 0.819 |
| 20 min | 0.909 | **0.935** | **0.935** | **0.935** | **0.836** | 0.824 | 0.824 | 0.824 |
| 30 min | 0.924 | **0.945** | **0.945** | **0.945** | 0.824 | **0.827** | **0.827** | **0.827** |
| **Zone 5** | | | | | | | | |
| 1 min | 0.779 | 0.854 | 0.405 | **0.896** | 0.411 | 0.775 | 0.353 | **0.882** |
| 2 min | 0.788 | 0.798 | 0.839 | **0.842** | 0.414 | 0.699 | **0.75** | 0.352 |
| 5 min | 0.748 | 0.77 | **0.796** | 0.777 | 0.559 | 0.367 | **0.744** | 0.359 |
| 10 min | 0.696 | 0.723 | **0.756** | 0.673 | 0.468 | **0.65** | 0.641 | 0.35 |
| 20 min | **0.631** | 0.426 | 0.61 | 0.629 | **0.474** | 0.348 | 0.343 | 0.368 |
| 30 min | **0.596** | 0.402 | 0.539 | 0.59 | 0.452 | 0.344 | **0.608** | 0.361 |
| **Zone 6** | | | | | | | | |
| 1 min | 0.872 | **0.961** | **0.96** | **0.96** | 0.895 | 0.963 | **0.96** | **0.96** |
| 2 min | 0.837 | **0.953** | 0.946 | 0.947 | 0.907 | **0.951** | 0.946 | **0.95** |
| 5 min | 0.872 | 0.915 | **0.939** | 0.933 | 0.823 | 0.915 | **0.934** | 0.929 |
| 10 min | 0.85 | 0.886 | **0.906** | 0.883 | 0.86 | 0.883 | **0.903** | 0.879 |
| 20 min | 0.72 | 0.863 | **0.875** | 0.815 | 0.707 | 0.848 | **0.873** | 0.864 |
| 30 min | **0.924** | 0.891 | 0.83 | 0.898 | 0.824 | 0.885 | 0.836 | **0.898** |
| **Zone 7** | | | | | | | | |
| 1 min | 0.756 | 0.921 | 0.912 | **0.932** | 0.854 | 0.965 | 0.968 | **0.97** |
| 2 min | 0.824 | 0.888 | 0.87 | 0.889 | 0.899 | 0.898 | **0.95** | 0.93 |
| 5 min | 0.783 | 0.82 | 0.862 | 0.809 | 0.833 | 0.875 | **0.938** | 0.817 |
| 10 min | 0.763 | 0.645 | 0.723 | **0.764** | 0.798 | 0.745 | **0.908** | 0.81 |
| 20 min | 0.667 | 0.608 | 0.695 | **0.724** | 0.834 | 0.775 | **0.877** | 0.779 |
| 30 min | 0.665 | 0.578 | 0.577 | **0.685** | 0.744 | 0.704 | **0.833** | 0.719 |

**Table 3.** The MSE evaluation for zone 1, 2, 4–7. Bold means better performance.

| Time Horizon | Week-1 | | | | Week-2 | | | |
|---|---|---|---|---|---|---|---|---|
| | **Zone 1** | | | | | | | |
| | DT | LSTM | MLP | OPTnet | DT | LSTM | MLP | OPTnet |
| 1 min | 0.182 | 0.051 | 0.314 | **0.049** | 0.146 | 0.039 | 0.084 | **0.033** |
| 2 min | 0.218 | **0.077** | 0.091 | 0.085 | 0.092 | 0.058 | **0.055** | 0.069 |
| 5 min | 0.207 | **0.179** | 0.188 | **0.18** | 0.139 | **0.08** | 0.089 | **0.081** |
| 10 min | 0.228 | 0.318 | 0.259 | **0.226** | 0.202 | 0.099 | 0.088 | **0.078** |
| 20 min | 0.299 | 0.372 | 0.378 | **0.284** | 0.175 | 0.176 | 0.233 | **0.143** |
| 30 min | 0.335 | 0.428 | 0.415 | **0.324** | **0.149** | 0.46 | 0.344 | 0.173 |
| | **Zone 2** | | | | | | | |
| 1 min | **0.0** | 0.202 | 0.496 | 0.249 | 0.223 | **0.135** | 0.146 | 0.137 |
| 2 min | **0.0** | 0.297 | 0.153 | 0.085 | 0.266 | 0.148 | **0.14** | 0.146 |
| 5 min | **0.0** | 0.321 | 0.186 | 0.227 | 0.228 | 0.171 | 0.166 | **0.147** |
| 10 min | **0.0** | 0.503 | 0.171 | 0.218 | 0.246 | **0.148** | 0.182 | **0.148** |
| 20 min | **0.016** | 0.502 | 0.213 | 0.286 | 0.264 | **0.151** | 0.161 | **0.151** |
| 30 min | **0.047** | 0.501 | 0.197 | 0.222 | 0.267 | **0.153** | 0.182 | **0.153** |
| | **Zone 4** | | | | | | | |
| 1 min | 0.107 | **0.081** | **0.081** | **0.081** | 0.282 | 0.184 | 0.186 | **0.1** |
| 2 min | 0.088 | **0.08** | **0.08** | 0.048 | 0.226 | 0.185 | 0.185 | **0.163** |
| 5 min | 0.101 | **0.062** | 0.077 | 0.77 | **0.136** | 0.186 | 0.145 | 0.183 |
| 10 min | 0.093 | **0.073** | **0.073** | **0.073** | 0.179 | **0.181** | **0.181** | **0.181** |
| 20 min | 0.091 | **0.065** | **0.065** | **0.065** | **0.164** | 0.176 | 0.176 | 0.176 |
| 30 min | 0.076 | **0.055** | **0.055** | **0.055** | 0.176 | **0.173** | **0.173** | **0.173** |
| | **Zone 5** | | | | | | | |
| 1 min | 0.221 | 0.146 | 0.595 | **0.104** | 0.589 | 0.225 | 0.647 | **0.118** |
| 2 min | 0.212 | 0.202 | 0.161 | **0.158** | 0.586 | 0.301 | **0.25** | 0.648 |
| 5 min | 0.252 | 0.23 | **0.204** | 0.223 | 0.441 | 0.633 | **0.256** | 0.641 |
| 10 min | 0.304 | 0.277 | **0.244** | 0.327 | 0.532 | 0.65 | 0.359 | **0.35** |
| 20 min | **0.369** | 0.574 | 0.39 | 0.371 | **0.526** | 0.652 | 0.657 | 0.632 |
| 30 min | **0.404** | 0.598 | 0.461 | 0.41 | 0.548 | 0.656 | **0.392** | 0.639 |
| | **Zone 6** | | | | | | | |
| 1 min | 0.128 | **0.039** | 0.04 | 0.04 | 0.105 | **0.037** | 0.04 | 0.04 |
| 2 min | 0.163 | **0.047** | 0.054 | 0.053 | 0.093 | **0.049** | 0.054 | 0.05 |
| 5 min | 0.128 | **0.061** | 0.204 | 0.067 | 0.177 | 0.085 | **0.066** | 0.071 |
| 10 min | 0.15 | 0.114 | **0.094** | 0.117 | 0.14 | 0.117 | **0.097** | 0.121 |
| 20 min | 0.255 | 0.137 | **0.125** | 0.185 | 0.216 | 0.152 | **0.127** | 0.136 |
| 30 min | 0.28 | 0.109 | 0.17 | **0.102** | 0.293 | 0.115 | 0.164 | **0.102** |
| | **Zone 7** | | | | | | | |
| 1 min | 0.244 | 0.079 | 0.088 | **0.068** | 0.146 | 0.035 | 0.0032 | **0.03** |
| 2 min | 0.176 | 0.112 | 0.13 | **0.111** | 0.101 | 0.102 | **0.05** | 0.07 |
| 5 min | 0.217 | 0.18 | **0.138** | 0.191 | 0.167 | 0.125 | **0.062** | 0.183 |
| 10 min | 0.237 | 0.355 | 0.277 | **0.236** | 0.202 | 0.255 | **0.092** | 0.19 |
| 20 min | 0.333 | 0.392 | 0.305 | **0.276** | 0.166 | 0.225 | **0.123** | 0.221 |
| 30 min | 0.335 | 0.422 | 0.423 | **0.315** | 0.256 | 0.296 | **0.167** | 0.281 |

## 5. Conclusions

Buildings are significant contributors to global energy consumption, accounting for approximately 40% of the total, and they are responsible for about 36% of carbon emissions. Achieving occupant-centric control is crucial for zero emissions and decarbonization efforts. This paper has addressed these challenges by introducing OPTnet, an innovative occupancy prediction framework. OPTnet utilizes data from multiple sensors, including building occupancy, indoor environmental conditions, and HVAC operations, to forecast future occupancy presence in multiple zones. Through experimental analysis and comparisons with other prediction methods, such as DT, LSTM, and MLP, OPTnet demonstrated superior performance across different time horizons (1, 2, 3, 5, 10, 20, and 30 min) using a practical two-week dataset. The promising results obtained from the OPTnet method underscore its potential to significantly improve HVAC control systems and energy optimization strategies in buildings. By accurately predicting occupancy patterns, the OPTnet-based approach can lead to more efficient building management, ultimately resulting in substantial reductions in energy consumption and environmental impact. However, further validation and testing of the OPTnet framework in real-world buildings of varying sizes, types, and locations will be essential to assess its scalability and generalizability.

**Author Contributions:** Conceptualization, I.Q.; Methodology, I.Q. and K.S.; Software, K.S.; Validation, K.S.; Investigation, I.Q.; Resources, H.Y.; Data curation, I.Q., T.X. and H.Y.; Writing—original draft, I.Q. and K.S.; Writing—review & editing, I.Q., Q.Z. and T.X.; Visualization, I.Q. and K.S.; Supervision, K.S. and Q.Z.; Funding acquisition, Q.Z. All authors have read and agreed to the published version of the manuscript.

**Funding:** This work is supported by the National Natural Science Foundation of China under Grant No. 62192751 and 61425027, in part, by the Key R&D Project of China under Grant No. 2017YFC0704100, 2016YFB0901900, by the 111 International Collaboration Program of China under Grant No. BP2018006, the 2019 Major Science and Technology Program for the Strategic Emerging Industries of Fuzhou under Grant No. 2019-Z-1, and, in part, by the BNRist Program under Grant No. BNR2019TD01009, and the National Innovation Center of High Speed Train R&D project (CX/KJ-2020-0006).

**Data Availability Statement:** To promote transparency and reproducibility, we make the code publicly available at https://github.com/kailaisun/occupancy-prediction-binary.

**Conflicts of Interest:** The authors declare no conflict of interest.

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
