# Peer review of "Multi-Sensor-Based Occupancy Prediction in a Multi-Zone Office Building with Transformer"

_buildings, doi:10.3390/buildings13082002_

Round 1

Reviewer 1 Report

The authors introduce an Occupancy Prediction Transformer  network (OPTnet) for building occupancy prediction. It can robustly predict occupancy  presence in diverse rooms and time horizons. The authors fuse and feed multi-sensor data into a Transformer model to obtain the future occupancy presence in multiple zones. The authors provide  experimental analysis and comparison between existing occupancy prediction methods  and diverse time horizons. The main contributions of this paper are as follows: 

• The authorsintroduce OPTnet, a Transformer-based multi-sensor building occupancy prediction network to learn an effective fused representation. 

• To predict accurate occupancy, the authors process two-week real operating sensor data from a multi-zone office building, including building occupancy, indoor environmental  conditions, and HVAC operations.

• Through experimental analysis and comparison, the authors found that the OPTnet method outperformed existing algorithms (e.g., Decision Tree (DT), Long Short-Term Memorynetworks (LSTM), Multi-Layer Perceptron (MLP)).

• Considering the long or short occupancy prediction applications, the authors provide a comprehensive analysis and comparison of diverse time horizons to highlight the importance  of choosing the suitable time horizon.

The paper is well done but I have some fremarks:

The authors did not consider these importante scientific paper:

https://doi.org/10.3390/buildings13071675

https://doi.org/10.1016/j.buildenv.2021.108109

DOI 10.4081/jae.2019.947

doi: 10.3390/buildings8070083

- The authors should improve the conclusions

It is good

Author Response

Many thanks for your general comments and valuable suggestions

Reviewer 2 Report

The paper, based on the measured data, compares the accuracy of the prediction results of DT, LSTM, MLP and OPTnet and proves that the OPTnet mostly outperforms other machine learning algorithms among the diverse duty ratios. However, the article only proves that OPTnet algorithm has high accuracy. The content depth is still room for further refinement, such as will parameter adjustment affect the accuracy of OPTnet algorithm? Will changes in the amount of data and information collected affect the results?

Other recommendations are as follows:

[1] It is suggested to add vertical columns in table 1 to explain the use of each device in the experiment or the parameters tested.

[2] Line 239, other parameters that affect the thermal comfort of users include airflow speed, average radiation temperature, etc. Are they taken into account when collecting data?

[3] In section 3.3, in order to ensure accuracy, different algorithms may have different parameter Settings for the same problem. Is it reasonable to set the parameters of different algorithms uniformly and then compare the accuracy?

Minor editing of English language required.

Author Response

(The authors gave the same response as above.)

Round 2

Reviewer 2 Report

  • The author has explained the issues raised by the review comments.